# Anxiety among secondary school students in the war-torn Tigray, Ethiopia, 2024: A call for action

Haftom Tesfay Gebremedhin[1]*, Abadi Kidanemariam Berhe[2,3], Yemane Gebremariam Gebre[4], Alem Gebremariam[2,3], Mulu Ftwi Beraki[5], Tesfay Gebreslassie Gebrehiwot[6], Guesh Teklu Woldemariam[6], Embay Amare Alemseged[2], Haileslassie Tesfay Tadese[6], Yemane Berhane Tesfau[2]

**1** Department of Psychiatry, College of Medicine and Health Sciences, Adigrat University, Adigrat, Tigray, Ethiopia, **2** Department of Public Health, College of Medicine and Health Sciences, Adigrat University, Adigrat, Tigray, Ethiopia, **3** Tigray Health Research Institute, Mekelle, Tigray, Ethiopia, **4** Department of Pediatrics, College of Medicine and Health Sciences, Adigrat University, Adigrat, Tigray, Ethiopia, **5** Department of Midwifery, College of Medicine and Health Sciences, Adigrat University, Adigrat, Tigray, Ethiopia, **6** Department of Nursing, College of Medicine and Health Sciences, Adigrat University, Adigrat, Tigray, Ethiopia

\* haftomtesfay30@gmail.com

## Abstract

Adolescents are especially vulnerable to stress and trauma. Exposure to armed conflict significantly raises the risk of anxiety, which often lasts into the post-conflict phase. The war in northern Ethiopia has caused widespread trauma, displacement, and destruction of services. Understanding mental health after conflict is crucial for guiding recovery efforts, informing school-based programs, and shaping local public health priorities. However, data on the prevalence of post-war anxiety among secondary school students in the region is lacking. This study assessed the prevalence and factors associated with clinically significant anxiety among secondary school students in Tigray, Ethiopia. A school-based cross-sectional study was conducted among 608 randomly selected secondary school students in Adigrat Town, Tigray, Ethiopia. Data was collected using a structured self-administered questionnaire, including the Generalized Anxiety Disorder (GAD)-7 scale for anxiety assessment. Multivariable binary logistic regression was used to identify factors associated with clinically significant anxiety symptoms. The prevalence of clinically significant anxiety symptoms among students was 32.7% (95% CI: 28.9%, 36.5%). Among sociodemographic factors, being female (AOR = 6.52, 95% CI: 3.70, 11.46) and age ≥ 18 years (AOR = 3.19, 95% CI: 1.60, 6.39) were significantly associated with anxiety symptoms. Trauma-related experiences, including combat or exposure to a battlefield (AOR = 5.05, 95% CI: 1.95, 13.12), physical violence (AOR = 2.25, 95% CI: 1.32, 3.82), bullying (AOR = 2.26, 95% CI: 1.24, 4.11), and contact sexual abuse (AOR = 3.97, 95% CI: 1.58, 9.96), were significantly associated with anxiety symptoms. Suicidal ideation (AOR = 4.01, 95% CI: 1.96, 8.24) and depression (AOR = 3.99, 95% CI: 2.32, 6.87) were also significantly associated with anxiety symptoms. This study reveals a high prevalence of clinically

**Data availability statement:** All relevant data are within the paper and its Supporting information files.

**Funding:** The authors received no specific funding for this work.

**Competing interests:** The authors have declared that no competing interests exist.

significant anxiety symptoms among secondary school students, associated with gender, age, trauma exposure, and comorbid conditions. These findings highlight the need for school-based mental health screening and trauma-informed support to reduce long-term psychological effects in this vulnerable generation.

## Introduction

Armed conflict is a serious public issue, with more than a billion young people living in affected areas, predominantly in lower- and middle-income countries where 90% of the world's children and adolescents reside [1,2]. One in four young people globally is affected by armed conflict or disaster, often with little to no access to psychological support [3]. During post-conflict or war, an estimated one in five people has mental problems [4]. Children and youths exposed to conflict face a heightened risk of developing mental health problems, especially anxiety, depression, and post-traumatic stress disorder (PTSD) [5]. Anxiety is the hallmark of anxiety disorders and is an unmanaged, pervasive, unpleasant, and enduring state of negative emotion, marked by apprehensive anticipation of unforeseen and unavoidable future threats, accompanied by physical tension and heightened alertness [6]. Anxiety disorders are among the most common and impairing conditions in adolescence, often emerging early in life and predicting poorer educational performance, reduced general functioning, and long-term mental health difficulties [7–9].

Multiple factors contribute to the development of anxiety. Individual traits and exposure to stressful life events increase susceptibility to anxiety [10,11]. Trauma exposure, in particular, can lead to long-term mental health issues by causing physiological changes that create individual vulnerabilities to various mental health problems later in life [12]. As a result, prolonged exposure to stressors associated with conflict not only causes anxiety immediately but also significantly increases the risk of developing later PTSD [13]. Evidence indicates that adverse experiences can become biologically embedded; for instance, altered amygdala connectivity in children exposed to stress predicts future anxiety symptoms [14,15]. Early life stress is also associated with hyperactive hypothalamic-pituitary-adrenal (HPA) axis responses, which are connected to later anxiety symptoms [16,17]. This is supported by neurobiological research indicating that trauma-related changes in this system increase stress reactivity and cause ongoing anxiety [18].

Studies from populations affected by conflict indicate that anxiety symptoms frequently worsen over time. A prospective study of children and adolescents exposed to war found that anxiety symptoms increased over time. [19]. Similarly, among Syrian and Iraqi refugee youth resettled in the US, anxiety prevalence was 38% upon arrival and worsened over two years [20]. Globally, the prevalence of anxiety among adolescents during or after violence varies widely, from 23.7% to 94.9% [21–24]. In Ethiopia, a post-war study among high school students revealed a prevalence of 39.7%, highlighting the possible burden of mental illness in areas affected by conflict [25].

In non-conflict settings of Ethiopia, adolescent anxiety levels are much lower. A systematic review among children and youth showed prevalences ranging from

0.5% to 23% [26], and an Ethiopian school-based study in a stable area found a prevalence of 25.05% among secondary school students [27]. Even in Tigray, the prevalence of anxiety before the conflict was 11.48%, showing typical levels before the war started. However, during the conflict, a phone survey found that anxiety prevalence increased to 34.43% [28]. This represents a threefold increase and highlights the severe psychological effects of the war on young people.

The Tigray region in northern Ethiopia faced severe and ongoing suffering during the two-year siege and war. The population endured profound trauma following the war [29], including the forced displacement of over 2 million people [30], the deliberate destruction of 70–80% of health facilities and targeted attacks on health workers [31,32], widespread gender-based violence [33], and a prolonged, complete blockade of communication, electricity, and other necessities [29]. These conditions create severe ecological and psychosocial stress that can increase the risk of anxiety among young people.

The development of anxiety is affected by different demographic, behavioral, and trauma-related factors. Studies have shown that a higher prevalence of anxiety was observed among females and individuals with lower socioeconomic status [34,35]. Moreover, studies reported that being female [22,25,36,37], age [22,38,39], being in a higher grade in school [40], smoking [41], alcohol use [27,42], witnessing the murder of family/friends [25], and depression [25,27,43] have a positive relationship with anxiety. Furthermore, individuals with suicidal ideation exhibit higher anxiety symptom severity compared to those without it [44]. Understanding these factors in post-conflict settings is crucial for creating effective prevention and intervention strategies.

To understand these complex relationships, this study relies on Bronfenbrenner's Ecological Systems Theory [45] and the Transactional Model of Stress and Coping [46]. According to Bronfenbrenner's framework, war affects an adolescent's surroundings on several levels, ranging from community disruption in the exosystem to direct trauma in the microsystem [45]. According to the Transactional Model, people's mental health outcomes are shaped by how they cognitively evaluate and manage stressors [46,47]. When taken as a whole, these theories clarify how ecological disruptions caused by war interact with individual vulnerabilities and coping mechanisms to affect the trajectories of post-conflict anxiety [47–50].

A high prevalence of anxiety among young people was identified in a phone survey conducted during the ongoing war. Still, no school-based assessment has been performed since the conflict ended. Little is known about the extent of anxiety and related factors among adolescents after the Pretoria Peace Agreement, despite the challenging humanitarian situation in Tigray. Therefore, essential questions remain: What is the current level of anxiety among secondary school students in Tigray after the war, and what factors are significantly associated with it?

This pre-registered cross-sectional study, conducted in 2024, aims to fill this gap by assessing the prevalence of clinically significant anxiety symptoms and its associated factors among secondary school students in the Tigray Region. By comparing data from before and during wartime, this research offers valuable insights for policymakers, mental health professionals, and school systems seeking to support adolescent recovery in one of the world's most severely affected conflict zones.

## Methods and materials

### Study design, setting, and period

A school-based cross-sectional study was conducted from November 1–30, 2024, in Adigrat Town, one of the zonal capitals of the Tigray Region in northern Ethiopia. The town has four public and one private secondary school.

### Study participants

All 10,714 students from public and private secondary schools (grades 9–12) enrolled in the 2024–2025 academic year, comprising 4,671 males and 6,043 females, were included in the study population. Public schools accounted for 8,914 of these students. The age range of participants extended up to 24 years. This shows the disruption to the regional education system caused by the war, which led to delayed school progression for many students.

This study included students who had lived in the region before the Pretoria Agreement (a permanent ceasefire) and were attending classes during data collection. Students who were seriously sick during the data collection were excluded. For this study, 'seriously sick' was operationally defined as a student who was visibly unable to provide meaningful assent or complete the questionnaire due to an acute condition like a high fever, vomiting, severe pain, or an active psychotic episode.

**Sample size and sampling technique**

The sample size was calculated for the target population of secondary school students in Adigrat Town using the single-proportion formula. The calculation was based on the following assumptions: an anxiety proportion of 39.7%, taken from a study of conflict-affected school adolescents in Woldia Town [25]; a 95% confidence interval; a 5% margin of error; a 10% non-response rate; and a design effect of 1.5. Accordingly, the total sample size for this study was 608 study.

A stratified simple random sampling design with proportional allocation was employed. First, the schools were stratified into public and private secondary schools, and the sample size was proportionally allocated based on each school's student enrollment. Within each school type, students were further stratified by academic grade into 9th, 10th, 11th, and 12th grades, and proportional allocation was again used to ensure that the probability of selection remained equal across all strata.

Each stratum used complete and up-to-date student rosters with distinct identification numbers as the sampling frame. Using a computer-generated series of random numbers applied to the student identification numbers, the predefined number of participants for each grade-level stratum was chosen from these complete lists. Students were selected individually rather than in classroom clusters, and no volunteers were requested. The randomly selected participants were then invited to a comfortable setting, and they completed the self-administered questionnaires after an orientation.

**Data collection and instruments**

Four health professionals with master's degrees were adequately oriented to study participants to complete the structured, self-administered questionnaire, which comprised four sections. The first section assessed participants' sociodemographic characteristics. The second part of the questionnaire was the General Anxiety Disorder-7 (GAD-7). The use of this tool is well supported by its psychometric validation in Ethiopia. The GAD-7 has shown strong internal consistency (Cronbach's α = 0.77, McDonald's Omega = 0.77, 0.78). It has a unidimensional factor structure confirmed by factor analyses (CFI = 1.000, GFI = 1.000, RMSEA = 0.037) and established convergent and divergent validity. Additionally, it has shown consistent item analysis and measurement invariance across gender. This confirms its suitability for use with young populations in Ethiopia [51]. The tool has seven items, and each item is rated on a four-point Likert scale that ranges from 0, "not at all," to 3, "nearly every day." The scores of all items were summed to yield a total score ranging from 0 to 21. Based on the total score, anxiety symptom severity was categorized as minimal (0–4), mild [5–9], moderate [10–14], and severe [15–21,52,53]. In accordance with the standard international and Ethiopian validation cut-offs, a GAD-7 score of 10 or higher indicates the presence of clinically significant anxiety symptoms, while scores below 10 indicate no anxiety [51,52]. The current study found a Cronbach's alpha coefficient of 0.917, indicating excellent internal consistency.

The third part of the questionnaire assessed behavioral factors. Substance use, such as khat, tobacco, and alcohol, was evaluated using a yes-or-no question adapted from the Alcohol, Smoking, and Substance Involvement Screening Test (ASSIST) tool [54,55], a globally validated instrument with strong psychometric properties. It showed good internal consistency with Cronbach's α values ranging from 0.77 to 0.94 across substance categories and 0.89 for the Total Substance Involvement score [56]. Physical activity was assessed by a single question to determine whether the participants met the WHO-recommended level or not [57].

The final segment of the questionnaire was regarding psychosocial factors, which included sexual abuse, suicidality, bullying experiences, trauma exposure, depression, and social support. Lifetime exposure to sexual abuse was measured

with four questions from the ISPCAN Child Abuse Screening Tool Children's Version [58]. Suicidal ideation and attempts were assessed using questions adapted from the World Mental Health Composite International Diagnostic Interview (WMH-CIDI) [59]. The adapted questions had "Yes/No" answers and were self-administered. Bullying experiences were assessed by a single-item measure adapted from the WHO/CDC Global School-based Health Survey (GSHS) [60]. Trauma exposure histories were assessed using a tool question adapted from the PCL-5 Life Events Checklist, designed to identify events meeting DSM-5 Criterion A [61]. Participants answered "Yes" or "No" to a list of event types. Depression symptoms were assessed using the 9-item Patient Health Questionnaire (PHQ-9), a validated scale scoring each item from 0 ("not at all") to 3 ("nearly every day"), with total scores ranging from 0 to 27 [62]. A cut-off score of ≥10 indicated the presence of likely clinically significant depressive symptoms [63]. In our sample, the PHQ-9 demonstrated good reliability (Cronbach's α = 0.88). Social support was assessed using the Oslo-3 Social Support Scale (OSS-3), yielding a sum score ranging from 3 to 14 [64]. The total score from the OSSS-3 was categorized as follows: 3–8 indicated poor social support, 9–11 indicated moderate social support, and 12–14 indicated strong social support [65].

## Operational definition

**Anxiety:** Participants who scored 10 or higher out of 21 on the GAD-7 were considered to have anxiety, while those who scored below 10 were classified as not having clinically significant anxiety symptoms (moderate to severe) [52].

**Depression:** Participants who scored ≥10 out of 27 on the PHQ-9 were classified as having clinically significant depressive symptoms (moderate to severe) [63].

**Bullying**: Being bullied refers to a student's exposure to repeated incidents of bullying in the last 30 days. Bullying is defined as intentional and repeated aggression, including verbal (e.g., teasing, threats), physical (e.g., hitting, kicking), or social (e.g., exclusion) acts perpetrated by one or more students. It excludes mutual conflicts between peers of similar power or playful, consensual teasing [66,67].

## Data quality control

The Tigrigna language questionnaire was used to collect data. To ensure data quality, the original English questionnaire was translated into Tigrigna (the local language) through a rigorous process to ensure validity across cultures. First, two forward translations were combined into one draft. Then this draft was back-translated into English to ensure it matched the original concepts. An expert review panel comprising bilingual mental health professionals and a language expert assessed all versions for meaning, idioms, and ideas. Finally, the Tigrigna questionnaire was pretested two weeks before formal data collection among 43 students from non-participating schools, which confirmed that participants understood it well and that its internal reliability was acceptable (Cronbach's α of 0.97).

Supervisors and data collectors completed a comprehensive two-day training program. During fieldwork, on-site supervision was implemented at all data collection points. Daily, the supervisors checked the completeness of the questionnaires.

## Data analysis

The data were checked, coded, and entered into Epi-Data software (Version 3.1), then exported to SPSS (Statistical Package for the Social Sciences, Version 25) for data analysis. Descriptive statistics (percentages, means, standard deviations) were used to characterize respondents' socio-demographic attributes and other variables.

Binary logistic regression was used to determine the association between the independent variables and the outcome variable. Bivariate logistic regression was first performed to identify the associations between each independent variable and the outcome variable. Variables significant at p ≤ 0.25 in bivariate analyses were entered into multivariate logistic

regression to adjust for confounding effects. Statistical significance was defined as $p < 0.05$, with adjusted odds ratios (AORs) and 95% confidence intervals (CIs) computed. The results are presented in tables, which display frequency distributions and summary statistics (e.g., means and percentages), as well as in supplementary figures.

We fitted a multivariable logistic regression model that included only significant variables from the bivariate analyses ($p < 0.05$) to evaluate the robustness of our results. The results are robust to changes in model specification because the direction and strength of the associations in this parsimonious model were consistent with those in the main model.

The model's goodness of fit was assessed using the Hosmer-Lemeshow test, and the results indicated adequate fit ($P = 0.273$). The model showed significant explanatory power, with a Nagelkerke $R^2$ value of 0.578, and exceptional predictive accuracy, with an Area Under the Curve (AUC) of 0.905 (95% CI [0.88, 0.930], $p < .001$). Multicollinearity was also evaluated using variance inflation factors (VIFs), and all variables had VIFs below 5.

Interaction terms between sex and different trauma types were fitted in the model to examine effect modification. A statistically significant interaction of gender and contact sexual abuse was observed, which was further explored by stratified analyses. Two-tailed $p < 0.05$ was considered statistically significant for main effects, and the results were presented as Adjusted Odds Ratios (AORs) and 95% Confidence Intervals (CIs).

### Ethical considerations

Ethical clearance was obtained from Adigrat University, College of Medicine and Health Sciences, Health Research Ethics Review Committee, with the code number ADU.CMHS/HRERC/0009/2024. Additionally, a formal letter was received from the town Education Office, which was provided to the respective schools before the actual data collection.

Students were assured that their participation would not impact their academic performance, and confidentiality was strictly maintained. For students aged 18 and older, verbal informed consent was obtained. For students under 18, verbal informed consent was secured from their parents, and the students provided assent. Students were instructed to consult the trained and qualified facilitator for any study-related concerns. Privacy was maintained through spaced seating, preventing participants from viewing each other's questionnaire responses.

## Results

### Socio-demographic characteristics of participants

All 608 students were assessed and found eligible based on the criteria; 599 completed the self-administered questionnaire, resulting in a response rate of 98.2%. Participants' ages ranged from 15 to 24 years, with a mean age of 17.61 years (±1.63).

Out of the total respondents, 506 (84.5%) attended public schools. Most participants, 574 (95.8%), identified as members of the Tigrian ethnicity, and 545 (91.0%) followed the Orthodox faith. Approximately 385 participants (64.3%) lived with both parents. More than one-third of participants, 233 (38.9%), were in grade 9, while 179 (29.9%) reported having more than three close friends (Table 1).

### Behavioral characteristics of the participants

Substance use and physical activity levels were assessed among 599 secondary students. The findings revealed that 88.1% of participants reported ever using alcoholic beverages, while only 6.5% and 4.8% had ever used khat and tobacco, respectively. Current alcohol use was reported by 44.7% of students, and in terms of physical activity, 54.4% of students did not meet the recommended levels (Table 2).

### Trauma exposure and psychosocial-related characteristics of the study participants

Among the 599 students surveyed, the most striking finding was the high prevalence of battlefield exposure: 84.1% reported combat experience or exposure to the battlefield. Physical violence was also widespread: 45.9% reported direct

**Table 1. Distribution of socio-demographic characteristics of secondary school students in Adigrat Town, 2024, N = 599.**

| Variables | Category | Frequency | Percent |
|---|---|---|---|
| Gender | Male | 255 | 42.6 |
| | Female | 344 | 57.4 |
| Age | 15-17 years | 288 | 48.1 |
| | 18-24 Years | 311 | 51.9 |
| Family structure (living with) | Both parents | 385 | 64.3 |
| | Single parent | 138 | 23.0 |
| | Others* | 76 | 12.7 |
| Ethnicity | Tigrian | 574 | 95.8 |
| | Others** | 25 | 4.2 |
| Religion | Orthodox Christianity | 545 | 91.0 |
| | Catholic | 43 | 7.2 |
| | Others*** | 11 | 1.9 |
| School type | Public | 506 | 84.5 |
| | Private | 93 | 15.5 |
| Grade | Nine | 233 | 38.9 |
| | Ten | 126 | 21.0 |
| | Eleven | 117 | 19.5 |
| | Twelve | 123 | 20.5 |
| Having close friends | None | 125 | 20.9 |
| | One | 127 | 21.2 |
| | Two | 168 | 28.0 |
| | Three or more | 179 | 29.9 |
| Residence | Urban | 475 | 79.3 |
| | Rural | 124 | 20.7 |

**Note:** *Grandparents, Siblings, Uncles, **Amhara, Oromia, ***Muslim, Protestant.

experiences, while 28.5% witnessed such events. Regarding sexual violence, 16.9% experienced non-contact sexual abuse, and 13.0% reported contact sexual abuse.

In our sample, 23.9% of participants said they had thought about suicide at some point in their lives. Additionally, 12.2% reported that they had attempted suicide at least once. In the preceding 12 months, 19.9% experienced suicidal ideation, and 9.7% had attempted suicide. Recent bullying (within the past 30 days) was reported by 20.0% of participants (Table 3). Additionally, 49.2% reported poor social support (Fig 1).

## Prevalence of clinically significant anxiety symptoms

The overall magnitude of clinically significant anxiety symptoms among the study participants was 32.7%, with a 95% CI (28.9%, 36.5%). The sample included 344 females (57.4%) and 255 males (42.6%). Consistent with this distribution, the magnitude of anxiety was significantly higher among females (47.4%, 95% CI: 42.1%, 52.7%) compared to males (12.9%, 95% CI: 8.8%, 17.1%). Furthermore, almost one-seventh of the students were experiencing severe levels of anxiety symptoms, as shown in Fig 2.

## Factors associated with clinically significant anxiety symptoms

Multivariable logistic regression revealed that being female, being 18 or older, experiencing attacks, shootings, or stabbings, exposure to war, being bullied, having a history of contact sexual abuse, suicidal ideation, and depression were significantly associated with clinically significant anxiety symptoms (Table 4).

**Table 2. Distribution of behavioral characteristics among secondary school students in Adigrat Town, 2024, N = 599.**

| Variables | Categories | Frequency | Percent |
|---|---|---|---|
| Alcoholic beverages | | | |
| Ever use history | Yes | 528 | 88.1 |
| | No | 71 | 11.9 |
| Current use history | Yes | 268 | 44.7 |
| | No | 331 | 55.3 |
| Khat | | | |
| Ever use history | Yes | 39 | 6.5 |
| | No | 560 | 93.5 |
| Current use history | Yes | 25 | 4.2 |
| | No | 574 | 95.8 |
| Tobacco products | | | |
| Ever use history | Yes | 29 | 4.8 |
| | No | 570 | 95.2 |
| Current use history | Yes | 18 | 3 |
| | No | 581 | 97 |
| Hashish | | | |
| Ever use history | Yes | 26 | 4.3 |
| | No | 573 | 95.7 |
| Current use history | Yes | 19 | 3.2 |
| | No | 580 | 96.8 |
| Meeting recommended physical activity | Yes | 273 | 45.6 |
| | No | 326 | 54.4 |

Specifically, female students were 6.52 times more likely to develop anxiety compared to male students (AOR = 6.52, 95% CI: 3.70, 11.46). Students aged 18 years and older were 3.19 times more likely to experience anxiety compared to those younger than 18 years (AOR = 3.19, 95% CI: 1.60, 6.39). Students who had a history of combat or exposure to a battlefield were five times more likely to develop anxiety than those students without a history of combat or exposure to a battlefield (AOR = 5.05, 95% CI: 1.95, 13.12). Experiencing physical violence such as attacks, shootings, or stabbings increased the odds of anxiety by 2.25 times (AOR = 2.25, 95% CI: 1.32, 3.82). Students who had experienced bullying were 2.26 times more likely to have anxiety (AOR = 2.26, 95% CI: 1.24, 4.11). The history of contact sexual abuse elevated the likelihood of anxiety by a factor of 3.97 times (AOR = 3.97, 95% CI: 1.58, 9.96). Anxiety was 4.01 times more common in students with a history of suicide ideation (AOR = 4.01, 95% CI: 1.96, 8.24). The odds of developing anxiety were 3.99 times greater among students with depression than those without (AOR = 3.99, 95% CI: 2.32, 6.87).

Furthermore, Table 5 below presents the findings from testing for effect modification between gender and different types of traumas. Gender and contact sexual abuse were found to interact significantly (AOR = 2.79, 95% CI [1.85, 4.19], p < 0.001). Other interactions between trauma and gender did not show statistical significance. Stratified analyses were performed to interpret the significant interaction. The relationship between anxiety and contact sexual abuse was statistically significant for females (AOR = 4.11, 95% CI [1.31, 12.89], p = 0.015), but not for males (AOR = 2.20, 95% CI [0.38, 12.84], p = 0.381), suggesting a gender-specific effect in which females were more susceptible to anxiety after this particular trauma.

## Discussion

The high prevalence of clinically significant anxiety symptoms identified in this study indicates that nearly one in three (32.7%, 95% CI (28.9%, 36.5%) Tigrian secondary school students suffer from anxiety, highlighting the enduring mental

**Table 3. Trauma exposure and psychosocial-related characteristics among secondary school students in Adigrat Town, 2024, N = 599.**

| Traumatic event variables | Category | Frequency | Percent |
|---|---|---|---|
| Experiencing the sudden death of a family member or someone close | Yes | 139 | 23.2 |
| | No | 460 | 76.8 |
| Non-contact sexual abuse | Yes | 101 | 16.9 |
| | No | 498 | 83.1 |
| Contact sexual abuse | Yes | 78 | 13.0 |
| | Yes | 521 | 87.0 |
| Experiencing physical violence | Yes | 275 | 45.9 |
| | No | 324 | 54.1 |
| Witnessing physical violence | Yes | 171 | 28.5 |
| | No | 428 | 71.5 |
| History of combat or exposure to a battlefield | Yes | 504 | 84.1 |
| | No | 95 | 15.9 |
| Lifetime suicidal ideation | Yes | 143 | 23.9 |
| | No | 456 | 76.1 |
| Suicidal ideation in the last 12 months | Yes | 119 | 19.9 |
| | No | 480 | 80.1 |
| A lifetime suicidal plan | Yes | 73 | 12.2 |
| | No | 526 | 87.8 |
| Suicidal plan in the last 12 months | Yes | 54 | 9.0 |
| Lifetime suicidal attempt | Yes | 74 | 12.2 |
| | No | 526 | 87.8 |
| Suicidal attempt in the last 12 months | Yes | 47 | 7.8 |
| | No | 552 | 92.2 |
| Bullied | Yes | 120 | 20.0 |
| | No | 479 | 80.0 |
| Depression | Yes | 443 | 74.0 |
| | No | 156 | 26.0 |

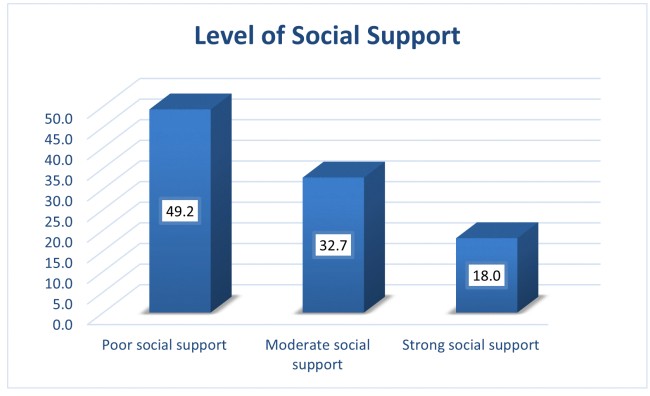

**Fig 1. Distributions of perceived social support levels among secondary school students in Adigrat Town, Ethiopia, 2024, N = 599.**

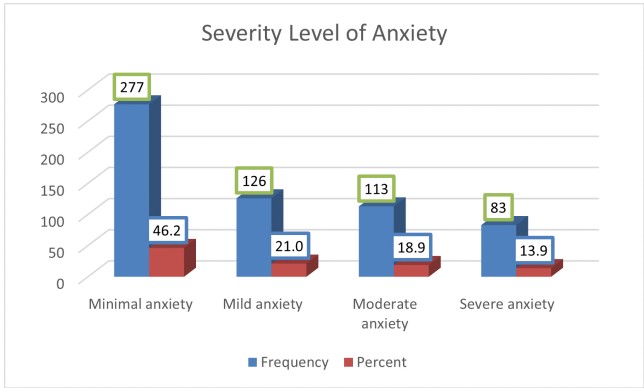

**Fig 2. Distribution of anxiety severity level among secondary school students in Adigrat Town, Ethiopia, 2024, N = 599.**

health burden long after active hostilities have ended. When placed in context, this prevalence is significantly elevated. In non-conflict settings of Ethiopia, a systematic review among children and youth showed a prevalence of 0.5% to 23% [26], and a school-based study showed a prevalence of 25.05% [27]. Even within Tigray, the pre-war anxiety prevalence was 11.48% [28]. The current prevalence closely matches the 34.43% reported in a wartime survey of Tigrayan youth [28], despite differences in methods and timing. This underlines the strong and lasting psychological effects of the conflict.

Our finding aligns with the longitudinal research in Sierra Leone, which showed that exposure to war-related trauma strongly predicts ongoing mental health issues in young people even years after the conflict ended [68]. Furthermore, our result is consistent with the study conducted among conflict-affected Palestinian adolescents, which reported a prevalence of 30.9% [69]. All things considered, these findings indicate that Tigrian students continue to experience severe anxiety symptoms despite the official ceasefire agreement, which highlights the urgent need for focused, sustained mental health interventions in schools affected by war.

However, the prevalence in this study was slightly higher than that reported in Uganda (26.6%) [70], Afghanistan (23.9%) [22], and another study from Palestinian (23.7%) [23], and lower than estimates from Sudan (50.8%) [71], Woldia Town, Ethiopia (39.7%) [25], the Gaza Strip (38%) [20], and Syrian and Iraqi refugee youth (37%) [72]. These differences among countries affected by war are not surprising and should be interpreted with caution. Factors such as the tools used for assessment, the timing of evaluations relative to exposure to conflict, the nature and severity of the conflicts, and various sociocultural factors likely explain much of the observed variation.

Female students were 6.52 times more likely to develop anxiety than male students in this study. This result is consistent with other studies conducted in Palestine [22], Ethiopia [25], and India [36]. The higher vulnerability could be due to a variety of factors that are more common in females, including their vulnerability to gender-related trauma like rape and violence [73,74], as well as socio-economic factors like gender disparities in caregiving and economic conditions [75]. Other pressures associated with school performance, friendships, and sleep problems could also be contributing to higher levels of anxiety in girls [76]. In addition, the lack of mental health care and the stigma associated with mental illness could also be contributing to delays in seeking care [77].

Students aged 18 years and older were 3.19 times more likely to experience anxiety compared to those younger than 18 years. Other studies support this finding [22,39]. This could be better explained by the fact that the period of transition from adolescence to young adulthood is often associated with heightened anxiety levels as individuals navigate new challenges such as higher education, career choices, and financial independence [78,79].

Students who were exposed to traumatic events such as violent attacks or battlefields were much more likely to experience anxiety symptoms than those who were not exposed. This is consistent with the evidence showing that individuals

**Table 4. Factors associated with clinically significant anxiety symptoms among secondary school students in Adigrat, 2024, N = 599.**

| Variables | Category | Anxiety | | COR (95% CI) | AOR (95% CI) | P-value |
|---|---|---|---|---|---|---|
| | | Yes | No | | | |
| Gender | Male | 33 | 222 | 1 | 1 | |
| | Female | 163 | 181 | 6.06 (3.97, 9.24) | 6.52 (3.70, 11.46) | 0.000** |
| Age | 15-17 | 59 | 229 | 1 | 1 | |
| | 18-24 | 137 | 174 | 3.06 (2.13, 4.40) | 3.19 (1.60, 6.39) | 0.001** |
| Family structure | With both parents | 116 | 269 | 1 | 1 | |
| | With a single parent | 44 | 94 | 1.09 (0.71, 1.65) | 0.82 (0.45, 1.48) | 0.506 |
| | Others | 36 | 40 | 2.09 (1.27, 3.44) | 0.99 (0.48, 2.03) | 0.975 |
| Grade | Nine | 51 | 182 | 1 | 1 | |
| | Ten | 44 | 82 | 1.92 (1.19, 3.10) | 1.21 (0.58, 2.50) | 0.609 |
| | Eleven | 50 | 67 | 2.66 (1.65, 4.31) | 0.90 (0.38, 2.12) | 0.806 |
| | Twelve | 51 | 72 | 2.53 (1.57, 4.06) | 0.71 (0.29, 1.70) | 0.437 |
| Social support | Strong | 23 | 85 | 1 | 1 | |
| | Moderate | 61 | 135 | 1.67 (0.96, 2.90) | 0.98 (0.46, 2.07) | 0.955 |
| | Poor | 112 | 183 | 2.26 (1.35, 3.79) | 1.19 (0.57, 2.45) | 0.644 |
| Current alcohol use | No | 85 | 246 | 1 | 1 | |
| | Yes | 111 | 157 | 2.05 (1.45, 2.89) | 0.99 (0.60, 1.62) | 0.955 |
| Sudden death of a family member or someone close | No | 114 | 346 | 1 | 1 | |
| | Yes | 82 | 57 | 4.37 (2.93, 6.51) | 1.12 (0.62, 2.05) | 0.702 |
| Experiencing physical violence | No | 69 | 255 | 1 | 1 | |
| | Yes | 127 | 148 | 3.17 (2.22, 4.53) | 2.25 (1.32, 3.82) | 0.003* |
| Witnessing physical violence | No | 106 | 322 | 1 | 1 | |
| | Yes | 90 | 81 | 3.38 (2.33, 4.90) | 1.42 (0.81, 2.51) | 0.224 |
| History of combat or exposure to a battlefield | No | 6 | 89 | 1 | 1 | |
| | Yes | 190 | 314 | 8.98 (3.85, 20.92) | 5.05 (1.95, 13.12) | 0.001** |
| Bullying experience | No | 129 | 350 | 1 | 1 | |
| | Yes | 67 | 53 | 3.43 (2.27, 5.18) | 2.26 (1.24, 4.11) | 0.008* |
| Non-contact sexual abuse | No | 142 | 356 | 1 | 1 | |
| | Yes | 54 | 47 | 2.88 (1.86, 4.46) | 0.49 (0.21, 1.13) | 0.094 |
| Contact sexual abuse | No | 143 | 378 | 1 | 1 | |
| | Yes | 53 | 25 | 5.60 (3.36, 9.36) | 3.97 (1.58, 9.96) | 0.003 |
| History of suicide ideation | No | 96 | 360 | 1 | 1 | |
| | Yes | 100 | 43 | 8.72 (5.72, 13.31) | 4.01 (1.96, 8.24) | 0.000** |
| History of plans for committing suicide | No | 143 | 383 | 1 | 1 | |
| | Yes | 53 | 20 | 7.10 (4.10, 12.29) | 1.16 (0.44, 3.04) | 0.766 |
| History of attempted suicide | No | 142 | 383 | 1 | 1 | |
| | Yes | 54 | 20 | 7.28 (4.21, 12.60) | 0.89 (0.33, 2.39) | 0.823 |
| Depression | No | 111 | 45 | 1 | 1 | |
| | Yes | 85 | 358 | 10.39 (6.83, 15.80) | 3.99 (2.32, 6.87) | 0.000** |

**Note:** 1.00 remained for the reference category, *significance level at p-value <0.05, ** significance level at p-value <=0.001.

**Abbreviations:** COR, Crude Odds Ratio; AOR, Adjusted Odds Ratio; CI, confidence interval.

**Table 5. Interaction effect between gender and trauma types on anxiety among secondary school students in Adigrat, 2024, N = 599.**

| Interaction term | COR (95% CI) | AOR (95% CI) | P-value |
|---|---|---|---|
| Gender*Contact sexual abuse | 2.92 (2.13, 4.01) | 2.79 (1.85, 4.19) | 0.000 |
| Gender*non-contact sexual abuse | 1.92 (1.49, 2.47) | 1.10 (0.78, 1.55) | 0.604 |
| Gender*Sudden death of a family member or friend | 1.20 (1.02, 1.40) | 1.16 (0.90, 1.50) | 0.253 |
| Gender*Experiencing physical violence | 1.13 (0.96, 1.32) | 0.94 (0.74, 1.18) | 0.590 |
| Gender*Witnessing physical violence | 1.18 (1.01, 1.38) | 1.05 (0.81, 1.36) | 0.713 |
| Gender*History of combat or exposure to a battlefield | 0.293 (0.19, 0.44) | 1.15 (0.86, 1.54) | 0.355 |

**Note:** The model contained all the listed interaction terms and main effects. Male and female were coded as 0 and 1, respectively. A significant interaction suggests that the impact of the trauma on anxiety symptoms is different for men and women.

who were exposed to violent conflicts experience greater levels of anxiety symptoms due to direct exposure to traumatic events, including the witnessing of violence or being forced to relocate [20,25,80]. Therefore, early identification and appropriate mental health care, as well as encouraging positive ways of coping, are especially important in a post-conflict environment [69,81].

Bullying victimization and mental health outcomes are significantly correlated, as evidenced by the finding that students who experienced bullying were 2.26 times more likely to have anxiety. This finding is consistent with previous research showing that exposure to bullying can have adverse psychological effects, such as an increase in anxiety symptoms [82,83]. Bullying-induced chronic stress may dysregulate the HPA axis, increasing susceptibility to anxiety disorders [84], while growing evidence also suggests that bullying could change brain structures involved in emotional regulation, such as the amygdala [85]. Moreover, victims of bullying often experience social withdrawal and negative self-perception, which further exacerbates anxiety symptoms [86]. Collectively, these mechanisms highlight how bullying impacts mental health through both physiological and psychological pathways.

Anxiety was about four times more common among students who had experienced contact sexual abuse compared to those who had not. This strong link supports earlier research indicating that sexual abuse is a major risk factor for anxiety disorders, including panic disorder, generalized anxiety disorder, and PTSD [87]. Furthermore, this finding aligns with evidence that women who experience childhood sexual abuse are more likely to develop internalizing disorders, like anxiety [88]. Neurobiological evidence suggests that trauma can disrupt the HPA axis, increasing stress reactivity and maintaining anxiety [18], along with social and cognitive factors like hypervigilance, stigma, and lack of support [89,90], might increase this vulnerability. These results highlight the need for gender-sensitive, trauma-informed treatments, such as cognitive-behavioral therapy (CBT) and eye movement desensitization and reprocessing (EMDR). These approaches have shown success in reducing anxiety for abuse survivors [91].

Students who have thought about suicide are 4.01 times more likely to experience anxiety. This aligns with studies that show a strong link between anxiety disorders and suicidal thoughts [92,93]. Anxiety symptoms, especially worry and trouble sleeping, can predict suicidal thoughts on their own, even after considering depression [92]. There is also evidence of a relationship between the severity of anxiety and suicidal behaviors [93]. These findings indicate that anxiety and suicidal thoughts may influence each other. Therefore, it is important to screen students who report suicidal thoughts for anxiety and vice versa, since dealing with both may require combined treatment.

Students with depression are almost four times more likely to develop anxiety symptoms compared to those without depression symptoms. This supports the established connection between anxiety and depression, as shown in the tripartite model, which points out shared negative feelings and unique traits of depression and anxiety [94]. Previous research shows significant overlap between the two conditions [94,95]. Since having both conditions often leads to worse treatment

results and higher chances of relapse, it is crucial to identify overlapping symptoms early and focus on treatments that address common factors [93,94].

## Limitations and strengths of the study

The cross-sectional design of this study limits the ability to establish causal relationships between anxiety and its risk factors. However, this study provides significant preliminary insights for future longitudinal research.

The following aspects limit the generalizability of these findings. First, since the data were collected from secondary schools in Adigrat town, the results might not represent the whole Tigray region or other post-conflict areas. Second, the school-based sampling frame may limit the extent to which we can generalize our findings, as it leaves out-of-school youth who might be at greater risk for trauma exposure and anxiety. Third, the inclusion of students up to 24 years old may limit direct comparisons with studies done in typical high school populations. We should keep this potential selection bias in mind when interpreting the results.

Furthermore, self-reported data on trauma exposure (e.g., bullying, sexual abuse) may be influenced by recall or social desirability biases, leading to potential underreporting. To address this, in addition to using well-validated and standardized tools, participants' anonymity was ensured, which strengthens the reliability of the responses.

## Conclusion

Nearly one-third of high school students were found to have clinically significant anxiety symptoms. This anxiety was significantly associated with being female, age 18 or older, direct exposure to battlefield or combat situations, physical violence, bullying, contact sexual abuse, and co-occurring suicidal ideation and depression.

These findings highlight the urgent need for targeted mental health programs in schools. Such initiatives should go beyond general support to include early screening for these specific risk factors and provide effective, trauma-informed treatments. For example, trauma-focused cognitive behavioral therapy can help address the unique psychological issues arising from war exposure. It's also crucial to integrate effective suicide risk assessments and depression management. This approach is necessary to reduce the long-term psychological impact on this vulnerable generation.

## Supporting information

**S1 Table. Completed STROBE checklist for cross-sectional studies.**
(DOC)

**S1 Data. Data sets for anxiety among secondary school students in the war-torn Tigray (n = 599).**
(SAV)

## Acknowledgments

The authors would like to thank school directors and the study participants for their dedicated cooperation, which made the study possible.

## Author contributions

**Conceptualization:** Haftom Tesfay Gebremedhin, Abadi Kidanemariam Berhe, Yemane Gebremariam Gebre, Alem Gebremariam, Mulu Ftwi Beraki, Tesfay Gebreslassie Gebrehiwot, Guesh Teklu Woldemariam, Embay Amare Alemseged, Haileslassie Tesfay Tadese, Yemane Berhane Tesfau.

**Data curation:** Haftom Tesfay Gebremedhin, Abadi Kidanemariam Berhe, Alem Gebremariam, Mulu Ftwi Beraki, Tesfay Gebreslassie Gebrehiwot, Guesh Teklu Woldemariam, Haileslassie Tesfay Tadese, Yemane Berhane Tesfau.

**Formal analysis:** Haftom Tesfay Gebremedhin.

**Investigation:** Haftom Tesfay Gebremedhin, Abadi Kidanemariam Berhe, Alem Gebremariam, Yemane Berhane Tesfau.

**Methodology:** Haftom Tesfay Gebremedhin, Abadi Kidanemariam Berhe, Yemane Gebremariam Gebre, Alem Gebremariam, Mulu Ftwi Beraki, Tesfay Gebreslassie Gebrehiwot, Guesh Teklu Woldemariam, Embay Amare Alemseged, Haileslassie Tesfay Tadese.

**Project administration:** Haftom Tesfay Gebremedhin, Abadi Kidanemariam Berhe, Yemane Gebremariam Gebre, Yemane Berhane Tesfau.

**Resources:** Haftom Tesfay Gebremedhin, Yemane Gebremariam Gebre, Mulu Ftwi Beraki.

**Software:** Haftom Tesfay Gebremedhin.

**Supervision:** Haftom Tesfay Gebremedhin, Abadi Kidanemariam Berhe, Yemane Gebremariam Gebre, Alem Gebremariam, Mulu Ftwi Beraki, Tesfay Gebreslassie Gebrehiwot, Guesh Teklu Woldemariam, Embay Amare Alemseged, Haileslassie Tesfay Tadese, Yemane Berhane Tesfau.

**Validation:** Haftom Tesfay Gebremedhin, Abadi Kidanemariam Berhe, Alem Gebremariam, Yemane Berhane Tesfau.

**Visualization:** Abadi Kidanemariam Berhe, Yemane Gebremariam Gebre, Alem Gebremariam, Yemane Berhane Tesfau.

**Writing – original draft:** Haftom Tesfay Gebremedhin, Mulu Ftwi Beraki, Tesfay Gebreslassie Gebrehiwot, Guesh Teklu Woldemariam, Embay Amare Alemseged, Haileslassie Tesfay Tadese.

**Writing – review & editing:** Haftom Tesfay Gebremedhin, Abadi Kidanemariam Berhe, Yemane Gebremariam Gebre, Alem Gebremariam, Mulu Ftwi Beraki, Yemane Berhane Tesfau.

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
