## [Decision Letter · Decision Letter 0]

29 Oct 2025

PMEN-D-25-00280

Anxiety among secondary school students in the war-torn Tigray, Ethiopia, 2024: a call for action

PLOS Mental Health

Dear Dr. Gebremedhin,

Thank you for submitting your manuscript to PLOS Mental Health. After careful consideration, we feel that it has merit but does not fully meet PLOS Mental Health’s publication criteria as it currently stands. Therefore, we invite you to submit a revised version of the manuscript that addresses the points raised during the review process.

Please respond to all reviewers, and my editor comments (below). Include a point-by-point response with page numbers, referring to the revised version. Include STROBE observational studies checklist (with page numbers) and flow diagram where possible. Ensure the manuscript is clearly written, and all results agree across the abstract, tables and text. Avoid using Excel for figures.

**Comments from the PLOS editorial office**: Upon internal evaluation of the reviews provided, we kindly request you to disregard the reviewer report provided by Reviewer 4. No amendments are required in response to Reviewer 4’s comments.

We look forward to receiving your revised manuscript.

Kind regards,

Gareth Hagger-Johnson

Academic Editor

PLOS Mental Health

Journal Requirements:

Additional Editor Comments (if provided):

Please revise the manuscript, responding to comments from all reviewers below.

Please write in clear English, and ensure the abstract matches the tables and results. When these differ, it suggests an earlier version of the results has remained. Proofread and check everything carefully before resubmitting.

Please use point by point responses, with page numbers referring to the revised version.

Guidance:

https://journals.plos.org/ploscompbiol/article?id=10.1371/journal.pcbi.1005730

https://www.nature.com/documents/Effective_Response_To_Reviewers-1.pdf

https://pmc.ncbi.nlm.nih.gov/articles/PMC11377928/

The manuscript will require further peer review on resubmission.

EDITOR COMMENTS

L71 Is "controllable" the right word - most cognitive behavioural techniques are built on the assumption that thoughts are controllable and anxiety can be managed.

L84 It can be inferred in this paragraph that anxiety during the conflicts themselves occurs, and chronic exposure to the stressors associated with conflict is a risk fator for later PTSD. But this isn't actually stated explicitly.

L102 Is "higher proportion of anxiety" the right phrase, do you mean prevalence? Prevalence is not the same as developmental characteristics.

L115 Use "prevalence" rather than "rate" for consistency.

L122 Now we have "level" in addition to "rate" and "prevalence" - be consistent throughout.

L127 Rewrite the end of the introduction to state a clear research question, including any pre-registration and subgroup analyses (e.g. gender) proposed. Remove statements like "The findings..." which belong in the discussion (and avoid sounding journalistic). How does your question differ from the existing prevalence survey (34%) also done on youth? Not clear here.

L154 If this is not a simple random sample (you mention stratification which is not simple random sampling), you need to adjust for the complex survey design features in the analysis to obtain correct standard errors.

L163 More detail needed on measures e.g. internal consistency reliability and validity in this country/language. Group into exposures, outcomes, covariates - justification needed.

L200 Translation into another language does not necessarily establish reliability and validity of psychometric measures.

L208 I think the analysis is incorrect if there was stratification, and it should be done with complex samples feature in SPSS. Otherwise the standard errors are incorrect.

P14 Is orthodox a religion? It is usually a subtype of a number of different religions.

Table 3 - the traumatic events are not specific to conflict. Are these positioned as covariates, exposures or something else? The status of your variables needs to be clear throughout.

L268 Here and elsewhere you are reporting prevalence for things not positioned as outcomes. It gives the paper an exploratory feel, lacking focus.

L388 The point about persistent exposure to trauma and dysregulation of HPA axis is made well here, and needs more strongly stating in the introduction.

L432 Clinically significant levels of anxiety, rather than anxiety?

END OF EDITOR COMMENTS AND REQUESTS

Reviewers' comments:

Reviewer's Responses to Questions

**Comments to the Author**

1. Does this manuscript meet PLOS Mental Health’s publication criteria? Is the manuscript technically sound, and do the data support the conclusions? The manuscript must describe methodologically and ethically rigorous research with conclusions that are appropriately drawn based on the data presented.

Reviewer #1: Partly

Reviewer #2: Yes

Reviewer #3: Yes

Reviewer #4: Yes

2. Has the statistical analysis been performed appropriately and rigorously?

Reviewer #1: Yes

Reviewer #2: Yes

Reviewer #3: Yes

Reviewer #4: Yes

3. Have the authors made all data underlying the findings in their manuscript fully available (please refer to the Data Availability Statement at the start of the manuscript PDF file)?

Reviewer #1: No

Reviewer #2: Yes

Reviewer #3: Yes

Reviewer #4: Yes

4. Is the manuscript presented in an intelligible fashion and written in standard English?

Reviewer #1: No

Reviewer #2: No

Reviewer #3: Yes

Reviewer #4: Yes

5. Review Comments to the Author

Reviewer #1: Overall Assessment

This manuscript addresses the mental health consequences of armed conflict on adolescents in post-war Tigray, Ethiopia—an important and underexplored topic with significant public health implications. The study employs a cross-sectional design to examine anxiety prevalence and associated factors among 599 secondary school students. While the research addresses a critical gap in conflict-affected populations' mental health literature, several methodological and analytical concerns limit its contribution to the field.

Recommendation: MINOR REVISION

Strengths

1. Timely and Important Topic: The focus on post-conflict mental health in Ethiopian youth addresses a significant gap in the literature, particularly given the limited research on the aftermath of the Tigray conflict.

2. Robust Sample Size: With 599 participants and a 98.2% response rate, the study provides adequate statistical power for the analyses conducted.

3. Comprehensive Assessment: The inclusion of multiple validated instruments (GAD-7, PHQ-9, ASSIST, etc.) provides a thorough evaluation of anxiety and associated factors.

4. Strong Statistical Analysis: The use of multivariable logistic regression with appropriate model diagnostics (Hosmer-Lemeshow test, VIF assessment) demonstrates methodological rigor.

5. Practical Relevance: The findings have direct implications for mental health interventions in post-conflict educational settings.

Major Concerns

1. Theoretical Framework and Conceptualization

Critical Gap: The manuscript lacks a clear theoretical framework to guide the understanding of post-conflict anxiety development. While the authors cite relevant literature, they fail to articulate how war trauma specifically manifests in anxiety disorders among adolescents or what theoretical model underlies their variable selection.

Recommendation: Incorporate established trauma and resilience theories (e.g., ecological systems theory, stress and coping models) to provide theoretical grounding for the study design and interpretation of findings.

2. Methodological Limitations

Sampling Concerns:

• The study is limited to one town (Adigrat), significantly limiting generalizability to the broader Tigray region or other post-conflict contexts

• The exclusion criteria mention "seriously sick" students but provide no operational definition

• No discussion of potential selection bias through school-based sampling (excluding out-of-school youth who may have higher trauma exposure)

Cross-sectional Design Limitations:

• The inability to establish temporal relationships between trauma exposure and anxiety symptoms is particularly problematic when studying post-conflict populations

• No baseline pre-conflict data for comparison, despite referencing a wartime study with similar prevalence rates

Cultural Adaptation:

• While the authors mention translation procedures, there is insufficient discussion of the cultural validity of Western-developed instruments (GAD-7, PHQ-9) in the Ethiopian context

• Limited consideration of how cultural expressions of distress might influence anxiety reporting

3. Statistical and Analytical Issues

Variable Classification:

• The dichotomization of continuous anxiety scores (GAD-7 ≥10) may result in loss of important information about anxiety severity gradations

• Several trauma variables appear to have small cell sizes that could affect model stability

Missing Analysis:

• No exploration of interaction effects between key variables (e.g., gender × trauma exposure)

• Limited discussion of effect sizes and clinical significance of associations

• No sensitivity analyses to test robustness of findings

Minor Concerns

1. Presentation and Clarity

Writing Quality: The manuscript contains numerous grammatical errors and awkward constructions that impede readability. Professional editing is recommended.

Table Formatting: Tables 4 particularly would benefit from better formatting to improve interpretability of results.

Figure Quality: The figures are adequate but could be enhanced with better legends and clearer visual presentation.

2. Literature Review and Discussion

Literature Integration: The discussion appropriately contextualizes findings within existing research but could benefit from deeper engagement with post-conflict mental health literature from sub-Saharan Africa.

Limitations Section: While present, the limitations discussion understates several key methodological concerns, particularly regarding generalizability and causal inference.

Specific Technical Comments

Methods Section

1. Line 150-153: The sample size calculation should specify the actual proportion used (39.7%) more clearly in the methods rather than just referencing another study.

2. Line 189-190: The operational definition of anxiety using GAD-7 ≥10 should be better justified, particularly given that this represents moderate-severe anxiety rather than any anxiety.

Results Section

3. Table 4: Consider reporting confidence intervals for all odds ratios, not just those that are statistically significant.

4. Line 277-281: The gender-specific prevalence rates (47.4% for females, 12.9% for males) are consistent with the overall rate and gender distribution in the sample.

Discussion Section

5. Lines 305-314: The comparison with wartime prevalence is intriguing but needs more careful interpretation given different methodologies and timing.

6. Lines 432-437: The conclusion appropriately calls for targeted interventions but could benefit from more specific, actionable recommendations based on the study's risk factor findings.

Recommendations for Revision

Required Revisions:

1. Strengthen theoretical framework: Add a clear theoretical model explaining post-conflict anxiety development

2. Enhance methodological transparency: Provide more detail on sampling procedures, exclusions, and potential biases

3. Improve statistical reporting: Include effect sizes, confidence intervals for all associations, and sensitivity analyses

4. Address cultural validity: Discuss the appropriateness of Western instruments in the Ethiopian context

5. Expand limitations discussion: Acknowledge generalizability constraints and causal inference limitations

Suggested Improvements:

1. Consider additional analyses: Explore interaction effects and dose-response relationships with trauma exposure

2. Enhance discussion: Integrate findings more deeply with post-conflict mental health theory

3. Improve presentation: Professional editing for grammar and clarity

4. Strengthen conclusions: Provide more specific, evidence-based recommendations for intervention

Minor Editorial Issues

Additional Technical Issue Identified: There is an inconsistency in the manuscript regarding contact sexual abuse odds ratios - the abstract reports AOR=3.09 (95% CI: 1.21, 7.91) while Table 4 and the results text show AOR=3.97 (95% CI: 1.58, 9.96). This discrepancy requires correction.

• Inconsistent citation formatting

• Figure legends could be more descriptive

• Some statistical reporting lacks precision (confidence intervals, exact p-values)

Significance and Impact

This study contributes valuable empirical data on an understudied population in a critical post-conflict period. The high prevalence of anxiety (32.7%) and the persistence of mental health impacts well after the ceasefire are concerning findings with important policy implications. The identification of specific risk factors (female gender, older age, trauma exposure, comorbid conditions) provides actionable targets for intervention.

However, the study's impact is limited by its cross-sectional design, single-site focus, and lack of theoretical depth. The findings, while important, represent a descriptive snapshot rather than a comprehensive understanding of post-conflict mental health dynamics.

Final Recommendation

This manuscript addresses an important and timely topic with adequate methodology and clear findings. While there are notable limitations in theoretical grounding and generalizability, the study provides valuable empirical data on a vulnerable population. With appropriate revisions addressing the methodological and theoretical concerns raised, this work would make a meaningful contribution to the post-conflict mental health literature.

The paper should be accepted following minor revision that addresses the theoretical framework, methodological transparency, and presentation issues outlined above.

Reviewer #2: This is an interesting and alarming manuscript. The findings are not particularly novel but are still interesting and worthy of publication.

I do have several concerns that need to be addressed prior to publication:

Method section:

1. There is not enough data regarding the questionnaires used (aside of the GAD-7). Details have to be added

2. Why was the score 10 on the GAD-7 used as the cut-off for "anxiety"? This needs to be explained given the centrality of this variable in the manuscript. Same goes for the cut-off score for "depression", though it is less critical.

3. I was surprised to read that the ages of the students in the study sample were up to 24. In many parts of the world individuals graduate high school at the ages of 17-19. This needs to be briefly addressed.

Discussion section:

In relation to the comparison of the study results regarding prevalence of anxiety to previous studies I am doubtful that a difference between studies of few percent is meaningful. I particularly point here to what is presented in lines 326-327.

Moreover, the segment of the discussion dealing with this comparison could be much more concise. And more generally, the discussion section as a whole is rather lengthy.

Is the manuscript presented in an intelligible fashion and written in standard English?

Though the manuscript is overall easily readable, there are some problems with grammar and overall quality of writing. The authors should have a professional overgoing their manuscript in order to correct grammar mistakes, use correct terms and avoid unnecessary repetitions and redundant sentences (such as the one in lines 254-255).

Reviewer #3: Reviewer’s comment

2)

Abstract

The study

Too generic: The background starts with a general statement:

"Anxiety is a common mental health issue that significantly affects the daily life of human beings."

– This is overly broad and somewhat generic. It doesn't yet set up a compelling case for why anxiety needs to be studied in this specific population and setting.

The transition to "particularly in conflict zones" is useful, but the connection between war, anxiety, and post-conflict settings needs much stronger development.

Lack of specificity: It mentions "conflict zones" and "post-conflict areas" without naming the specific context (Tigray) until the last line.

Missing significance: It doesn't tell us why this study is important — for public health, policy, or local recovery efforts.

Weak transition to objective: The jump from "limited evidence" to "this study aims..." is too abrupt and unsubstantiated.

The results are clearly presented. It would be much better if you could do grouping or prioritizing the most impactful predictors. Because. It feels slightly overwhelming to list 8 risk factors one after another — can feel like a data dump.

The conclusion : Vague: "high prevalence" — compared to what? Other studies? National estimates?

Could say how the findings can inform interventions (e.g., school-based screening, trauma services).

Introduction

I suggest reconsidering how the introduction opens. While defining anxiety has its place, starting with a textbook-style definition may not immediately engage the reader or highlight the significance of your study. Given your focus on a post-conflict region, it would be more impactful to begin by framing the broader issue — that war-affected populations, particularly adolescents, are at heightened risk of anxiety and other mental health problems. This would help set the stage for your study in Tigray and guide the reader more clearly toward your research objective.

Overall, the introduction contains a considerable amount of general information on anxiety — including definitions, global statistics, and clinical descriptions — but it takes too long to get to the specific problem this study aims to address. To make a stronger introduction, I suggest reworking the opening paragraphs to:

Begin by highlighting the broader public health concern: the mental health impact of war and conflict, especially on adolescents.

Narrow down to the specific context of Tigray, Ethiopia, and the ongoing post-conflict challenges faced by school-aged youth.

Clearly state the research gap, such as the limited evidence on anxiety prevalence and associated factors among adolescents in this region.

it would strengthen the justification for the study to clarify a few important points. First, since anxiety and depression frequently co-occur with PTSD in post-conflict settings, it would be helpful to explain how your study specifically addresses anxiety as distinct from PTSD, or whether comorbidity was considered in your assessment.

the paper mentions that anxiety prevalence has reportedly tripled in the post-conflict period (to around 34%), but it would be important to clarify what the baseline was and how this compares not just globally, but also within other regions of Ethiopia. For instance, how does the prevalence of anxiety in Tigray compare with non-conflict regions of Ethiopia? What makes Tigray’s situation uniquely deserving of focused study?

while some relevant comparisons and previous studies are mentioned in the discussion section, it would be more appropriate to bring these into the introduction. Doing so would provide readers with a more complete picture of the existing evidence — both globally and locally — and more clearly articulate the research gap your study is addressing.

While the traumatic exposures listed in the introduction (e.g., displacement, violence, health system collapse) certainly highlight the severity of the conflict, listing these alone doesn’t fully justify an anxiety-specific study. These exposures are commonly cited in research on PTSD or depression, and the way they're presented here seems to support a broader mental health burden rather than a focused anxiety investigation.

To better justify the study, I suggest explaining how these specific types of trauma are known to contribute to heightened anxiety — especially in adolescents — and why this deserves focused attention, independent of PTSD or depression.

Additionally, the manuscript mentions multiple times that the study is “unique,” but this needs to be clarified. If the uniqueness lies in the timing (post–Pretoria Peace Agreement), the population (secondary school students), or the focus on anxiety as a primary outcome, that should be explicitly stated and justified. Notably, many post-conflict studies treat anxiety as secondary to PTSD or depression — so if your study breaks from that pattern, this would be a strong and novel point worth highlighting more clearly in the introduction.

Reframing the introduction in these ways would help the reader immediately understand the relevance and urgency of your study and better align with the research objective.

Methods

Design setting and period …

This section includes some context that, while important in general, may not be directly relevant to the study's design, setting, and timeframe. If I correctly understand what you were saying, for example, mentioning the decrease in the number of functioning schools — possibly due to conflict-related damage — feels more like a journalistic narrative than a methodological description.

As a researcher, it’s important to maintain focus on the core components of your study design. I understand and sympathize with the broader challenges the region faces, but in this section, I would recommend keeping the reader’s attention on the information essential to understanding where, when, and how the study was conducted. Consider moving context about the state of the “education system” (usually system a nationwide-program in my understanding) to a background or discussion section if you feel it’s necessary to include.

This would help avoid distraction and maintain a more concise, scientific tone in the methodology section.

Participants:

I believe the numerical data of the population based on gender, if possible, including the school and grade information belongs here.

For better presentation, the inclusion and exclusion criteria need to be further elaborated with reasons to justify your decision.

Sample size and sampling technique

The description of participants as a "single population" could be misleading, as they appear to differ significantly across several dimensions — including gender, school type (private vs. public), and war-related experiences (combatants, victims…) because they were your important predictors. These subgroups may have very different exposure profiles or risk factors for anxiety. It would be helpful to clarify what is meant by "single population" in this context. If this refers only to the target group being "secondary school students in Adigrat," that should be clearly stated to avoid confusion.

It’s not fully clear how randomization was implemented at the stratum level. While the use of a computer-generated lottery is mentioned, it would help to explain how student identification numbers were obtained for each grade and school type (public/private), and how feasible it was to access and use this full list in practice. Was a complete list available for each stratum, or were students asked to volunteer and then selected randomly? Providing a bit more detail here would clarify the sampling procedure and reassure readers about the randomness of participant selection.

Instruments

In the instruments section, I recommend giving equal attention to the description of all tools used in the study. Currently, some instruments are briefly mentioned, while others (like the General Anxiety Disorder-7 (GAD-7) was described in more detail. It would improve clarity and consistency to follow a similar structure for each instrument — including source, number of items, scoring, and whether it was self- or interviewer-administered.

Additionally, the statement that suicidal ideation and attempts were assessed using questions adopted from the WMH-CIDI might be confusing. As far as I understand, the WMH-CIDI is a structured diagnostic interview typically administered by trained professionals, not self-administered. If you only used selected items from the CIDI within a self-administered questionnaire, it would be important to clarify that, so as not to give the impression that formal clinical diagnoses were made.

While the reported Cronbach's alpha of 0.97 for the GAD-7 indicates excellent internal consistency, such a high value may also suggest that some items are overly similar or redundant. This can sometimes occur when forward or backward translations result in items sounding too alike — particularly in languages with limited emotional vocabulary. It would be helpful to include a brief note in the discussion section addressing this possibility, to reassure readers that the high internal consistency does not compromise the scale’s validity or interpretability in your context.

Analysis: The current analysis approach is generally appropriate, but there are areas that could be improved. Relying on p-values from bivariate analysis to select variables for the multivariable model may overlook important confounders. It would be better to select variables based on theoretical relevance or a conceptual framework. Additionally, reporting more diagnostic measures (e.g., pseudo R², AUC) and considering interaction terms could strengthen the model. If anxiety severity levels were used, ordinal logistic regression might offer more nuanced insight. Finally, clarification on how missing data were handled would improve the transparency of the analysis.

Results

Well done!

Discussion

The reported anxiety prevalence (32.7%) is clearly concerning and deserves attention, but I would caution against characterizing it as exceptionally high without more context. The manuscript frequently refers to a “threefold increase” in anxiety during the war (34%) compared to pre-war levels, but no data or references are provided to support this claim — neither for pre-war Tigray nor for Ethiopia more broadly. This same comparative statement appears across several sections (introduction, methods, results, and now discussion), and yet no baseline data is cited.

If you are aware of existing studies or national surveys that report pre-conflict anxiety rates in Ethiopia, those should be included in the background section to help readers understand the magnitude and significance of your findings. If no such data exists, it would be better to either acknowledge the lack of baseline data or reframe the comparison more cautiously.

In the current form, this central comparison risks overstating the case and may mislead readers about the actual mental health trajectory before, during, and after the conflict.

Putting findings like in the Woldia Town study might give background if you put it in the introduction setting.

Reviewer #4: Thanks to all Authors.

The topic is highly relevant, timely, and contextually important, as the psychological effects of war on adolescents are often under-researched.

The focus on secondary school students in Tigray provides valuable insights into a vulnerable population.

The manuscript addresses a critical mental health concern in conflict settings, which adds to the global body of knowledge.

Comment on Clarity and Structure

Is the introduction well-structured and does it clearly present the problem?

The introduction sets the stage well, but it could more strongly highlight gaps in existing literature on war-related adolescent anxiety.

Consider streamlining the methodology section for clarity, especially around sample selection and data collection tools.

Methodological Feedback

It would be helpful to provide more detail on the anxiety measurement tool—whether it was validated for this cultural and linguistic context.

Clarify how confounding factors such as displacement, loss of family members, or lack of schooling were considered in the analysis.

Results and Discussion

The results are clearly presented, though adding more tables/figures could make patterns easier to understand.

The discussion is strong, but linking findings more explicitly to global literature on adolescents in conflict zones could strengthen the contribution.

Consider balancing the focus on risk factors with potential resilience factors among these students.

Language and Style

The manuscript is generally well-written, but minor grammatical revisions would improve readability.

Avoid overly technical jargon where possible—consider how practitioners or policymakers might use this information.

6. PLOS authors have the option to publish the peer review history of their article (what does this mean?). If published, this will include your full peer review and any attached files.

**Do you want your identity to be public for this peer review?** For information about this choice, including consent withdrawal, please see our Privacy Policy.

Reviewer #1: No

Reviewer #2: No

Reviewer #3:**Yes:**Amare Misganaw Mihret

Reviewer #4:**Yes:**Yihdego bites gebrezgi

Figure Resubmissions:

---

## [Decision Letter · Decision Letter 1]

10 Feb 2026

PMEN-D-25-00280R1

Anxiety among secondary school students in the war-torn Tigray, Ethiopia, 2024: a call for action

PLOS Mental Health

Dear Dr. Gebremedhin,

Thank you for submitting your manuscript to PLOS Mental Health. After careful consideration, we feel that it has merit but does not fully meet PLOS Mental Health’s publication criteria as it currently stands. Therefore, we invite you to submit a revised version of the manuscript that addresses the points raised during the review process.

The manuscript has been reassessed by a reviewer, and their comments are available below.

The reviewer raised a number of major concerns. They would like additional precision regarding terminology used for depression and anxiety. They also recommend clarifying the age of the participants in the text and range used to classify anxiety on the GAD scale. Recommendations have also been made to improve the discussion section. 

Could you please carefully revise the manuscript to address all comments raised?

We look forward to receiving your revised manuscript.

Kind regards,

Katherine Demi Kokkinias, Ph.D.

Staff Editor

PLOS Mental Health

Journal Requirements:

Additional Editor Comments (if provided):

Reviewers' comments:

Reviewer's Responses to Questions

**Comments to the Author**

1. If the authors have adequately addressed your comments raised in a previous round of review and you feel that this manuscript is now acceptable for publication, you may indicate that here to bypass the “Comments to the Author” section, enter your conflict of interest statement in the “Confidential to Editor” section, and submit your "Accept" recommendation.

Reviewer #2: (No Response)

2. Does this manuscript meet PLOS Mental Health’s publication criteria? Is the manuscript technically sound, and do the data support the conclusions? The manuscript must describe methodologically and ethically rigorous research with conclusions that are appropriately drawn based on the data presented.

Reviewer #2: Partly

3. Has the statistical analysis been performed appropriately and rigorously?

Reviewer #2: Yes

4. Have the authors made all data underlying the findings in their manuscript fully available (please refer to the Data Availability Statement at the start of the manuscript PDF file)?

Reviewer #2: (No Response)

5. Is the manuscript presented in an intelligible fashion and written in standard English?

Reviewer #2: Yes

6. Review Comments to the Author

Reviewer #2: The authors have made a noticeable effort to address my previous comments; however, several important issues still require further attention.

1. Data collection and instruments:

The description of the questionnaires is now more detailed, which is an improvement. Nevertheless, I take issue with the use of the term “anxiety” to describe participants who scored 10 or higher out of 21 on the GAD-7. Anxiety exists on a continuum, and using this terminology is imprecise. The authors should adopt more accurate language, such as “moderate to severe anxiety.” The same concern applies to the terminology used for “depression,”.

2. Age of participants:

In response to my previous comment regarding the age range of the participants, the authors state that “the broader age range in our sample, which includes students up to 24 years old, directly reflects the major disruptions to the education system caused by the war in the region.” This explanation is reasonable, but it must be explicitly addressed and discussed in the manuscript itself, as it has important implications for comparisons with high school student samples in most other countries.

3. Discussion section:

I remain unconvinced that the differences between the anxiety rates reported in the current study and those reported in previous studies (e.g., 26.6% in Uganda or 39.7% in Ethiopia) are sufficiently large to warrant an extensive and detailed discussion. In fact, it would have been more surprising had the rates been very similar. It should suffice to note that anxiety rates differ across war-affected countries, with some variation likely attributable to differences in methodology, characteristics of the conflicts, and contextual factors. This point could be adequately addressed in a single, concise paragraph.

More generally, the revised Discussion section remains overly long. For example, there is no need to elaborate on the correlation between anxiety and depression, which is a well-established finding in the literature.

7. PLOS authors have the option to publish the peer review history of their article (what does this mean?). If published, this will include your full peer review and any attached files.

**Do you want your identity to be public for this peer review?** For information about this choice, including consent withdrawal, please see our Privacy Policy.

Reviewer #2: No

Figure Resubmissions:

---

## [Decision Letter · Decision Letter 2]

25 Feb 2026

PMEN-D-25-00280R2

Anxiety among secondary school students in the war-torn Tigray, Ethiopia, 2024: a call for action

PLOS Mental Health

Dear Dr. Gebremedhin,

Thank you for submitting your manuscript to PLOS Mental Health. After careful consideration, we feel that it has merit but does not fully meet PLOS Mental Health’s publication criteria as it currently stands. Therefore, we invite you to submit a revised version of the manuscript that addresses the points raised during the review process.

I am pleased to say that I can now offer you a provisional acceptance for your submission. The referee has provided accept review and I have only one minor comment:

Methods. Sample size and sampling technique: “a total of 608 samples were used in this study” – should it possibly be “the total sample of 608 students was used in this study”, or “the total sample size for this study was 608”?

We look forward to receiving your revised manuscript.

Kind regards,

Helena R. Slobodskaya, M.D., Ph.D., D.Sc.

Academic Editor

PLOS Mental Health

Journal Requirements:

Additional Editor Comments (if provided):

Reviewers' comments:

Reviewer's Responses to Questions

**Comments to the Author**

1. If the authors have adequately addressed your comments raised in a previous round of review and you feel that this manuscript is now acceptable for publication, you may indicate that here to bypass the “Comments to the Author” section, enter your conflict of interest statement in the “Confidential to Editor” section, and submit your "Accept" recommendation.

Reviewer #2: All comments have been addressed

2. Does this manuscript meet PLOS Mental Health’s publication criteria? Is the manuscript technically sound, and do the data support the conclusions? The manuscript must describe methodologically and ethically rigorous research with conclusions that are appropriately drawn based on the data presented.

Reviewer #2: Yes

3. Has the statistical analysis been performed appropriately and rigorously?

Reviewer #2: Yes

4. Have the authors made all data underlying the findings in their manuscript fully available (please refer to the Data Availability Statement at the start of the manuscript PDF file)?

Reviewer #2: (No Response)

5. Is the manuscript presented in an intelligible fashion and written in standard English?

Reviewer #2: Yes

6. Review Comments to the Author

Reviewer #2: The authors responded well to all my comments and the manuscript is ready for publication

7. PLOS authors have the option to publish the peer review history of their article (what does this mean?). If published, this will include your full peer review and any attached files.

**Do you want your identity to be public for this peer review?** For information about this choice, including consent withdrawal, please see our Privacy Policy.

Reviewer #2: No

Figure Resubmissions:

---

## [Editor Report · Decision Letter 3]

4 Mar 2026

Anxiety among secondary school students in the war-torn Tigray, Ethiopia, 2024: a call for action

PMEN-D-25-00280R3

Dear Mr. Gebremedhin,

We are pleased to inform you that your manuscript 'Anxiety among secondary school students in the war-torn Tigray, Ethiopia, 2024: a call for action' has been provisionally accepted for publication in PLOS Mental Health.

Best regards,

Helena R. Slobodskaya, M.D., Ph.D., D.Sc.

Academic Editor

PLOS Mental Health